# Investigating Acute Hepatitis after SARS-CoV-2 Vaccination or Infection: A Genetic Case Series

**DOI:** 10.3390/biomedicines11102848

**Published:** 2023-10-20

**Authors:** Elisa Bernasconi, Matteo Biagi, Stefania Di Agostino, Carmela Cursaro, Cristina Felicani, Enrico Ronconi, Elena Franchi, Arianna Carmen Costanzo, Filippo Gabrielli, Alessia Cavicchioli, Giuseppe Ienopoli, Paolo Marenghi, Alessandra Bartoli, Beatrice Serra, Davide Scalabrini, Pamela Sighinolfi, Pietro Andreone

**Affiliations:** 1Department of Internal Medicine, Civil Hospital of Baggiovara, University of Modena and Reggio Emilia, Baggiovara, 41126 Modena, Italy; elisa.bernasconi94@gmail.com (E.B.); matteobiagi95@gmail.com (M.B.); stefania.diagostino@yahoo.it (S.D.A.); carmela.cursaro2022@libero.it (C.C.); cristinafelicani@gmail.com (C.F.); enrico.ronconi9@gmail.com (E.R.); elena.fr6@gmail.com (E.F.); ariannacostanzo23@icloud.com (A.C.C.); judeslash1@gmail.com (F.G.); alessia.cavicchioli.1990@gmail.com (A.C.); giuseppeienopoli10@gmail.com (G.I.); paolo.marenghi@libero.it (P.M.); alessandra.bartoli25@yahoo.com (A.B.); bea.s.93@hotmail.it (B.S.); dscalabrini@gmail.com (D.S.); sighinolfi.pamela@policlinico.mo.it (P.S.); 2Department of Internal Medicine, General, Emergency and Post-Acute, Division of Metabolic Internal Medicine, Civil Hospital of Baggiovara, Azienda Ospedaliero-Universitaria di Modena, Baggiovara, 41126 Modena, Italy

**Keywords:** autoimmune hepatitis, acute hepatitis, SARS-CoV-2 infection, COVID-19 vaccine, genetic background

## Abstract

(1) Background: Despite the advantages of COVID-19 vaccination, rare cases of acute hepatitis developing after the administration of the COVID-19 vaccine or the severe acute respiratory syndrome coronavirus 2 (SARS-CoV-2) infection have been reported. The aim of the study is to describe a case series of patients who experienced the onset of acute hepatitis, with or without autoimmune features, following SARS-CoV-2 vaccination or infection and to hypothesize a genetic susceptibility in the pathogenesis. (2) Methods: A group of patients with acute onset hepatitis following SARS-CoV-2 vaccination or infection were evaluated in our hepatology outpatient clinic, where they underwent biochemical and autoimmune tests. Hepatitis A (HAV), B (HBV), and C virus (HCV), cytomegalovirus (CMV), Epstein–Barr virus (EBV), and human immunodeficiency virus (HIV) infections were excluded. Patients with a diagnosis of autoimmune hepatitis (AIH) or drug-induced liver injury (DILI) underwent HLA typing and histological testing. (3) Results: Five patients experienced new-onset AIH after COVID-19 vaccination, one of which developed mild symptoms after vaccination that strongly worsened during subsequent SARS-CoV-2 infection. One patient had AIH relapse after COVID-19 vaccination while on maintenance immunosuppressive treatment. All of them had HLA DRB1 alleles known to confer susceptibility to AIH (HLA DRB1*03,*07,*13,*14), and in three of them, HLA DRB1*11 was also detected. Two patients developed acute hepatitis without autoimmune hallmarks which resolved spontaneously, both positive for HLA DRB1*11. (4) Conclusions: An association between AIH and COVID-19 vaccine or infection can be hypothesized in individuals with a genetic predisposition. In patients without autoimmune features and spontaneous improvement of hypertransaminasemia, the diagnosis of drug-induced liver injury (DILI) is probable. Further studies are needed to determine the presence of an actual association and identify a possible role of HLA DRB1*11 in the pathogenesis of acute liver injury after SARS-CoV2 vaccination or infection.

## 1. Introduction

In the recent literature, both the severe acute respiratory syndrome coronavirus 2 (SARS-CoV-2) infection and the COVID-19 vaccine have been hypothesized to be associated with the onset or relapse of autoimmune pathologies [1,2], including autoimmune or autoimmune-like hepatitis [3,4,5,6,7,8,9,10,11,12,13,14,15,16,17,18,19,20,21,22,23,24,25,26]. Autoimmune hepatitis (AIH) is a rare liver disease that affects mainly women and is characterized by circulating autoantibodies, hypergammaglobulinemia, and interface hepatitis on liver histology [27]. Although the annual incidence of AIH is around 2.08 cases per 100,000, the frequency of new cases is increasing by three percent every year [28], making the clinical burden of AIH high. The disease results from immune-mediated reactions against hepatocytes, triggered by an environmental risk factor, such as a drug or infection, in genetically predisposed individuals [29]. A strong genetic association has been identified with certain human leukocyte antigen (HLA) alleles [29].

A few cases of liver injury after SARS-CoV-2 infection [30,31,32,33,34] and COVID-19 vaccination [35,36], not presenting with an AIH pattern and resembling a drug-induced liver injury (DILI), have also been reported in the literature. DILI is a rare but sometimes severe adverse drug reaction that manifests with th eelevation of liver enzymes, with or without non-specific symptoms [37]. The pathogenesis of DILI involves a complex interaction between the drug or its metabolites and the host immune system, with a key role of restricted HLA associations in the activation of adaptive immune response [37]. Since both AIH and DILI are mediated by immunological reactions and their serological and histological findings are often similar [38], the differential diagnosis between these two conditions can be very challenging.

## 2. Patients and Methods

This is a descriptive study conducted among patients who presented at the Hepatology Unit of University Hospital of Baggiovara (Modena, Italy) for the evaluation of a sudden increase in liver enzymes, with or without symptoms, after the COVID-19 vaccine or SARS-CoV-2 infection. Prior to the exclusion of hepatitis A (HAV), B (HBV), and C virus (HCV), cytomegalovirus (CMV), Epstein–Barr virus (EBV), and human immunodeficiency virus (HIV) infections, eight patients were included in the present study. They were characterized under biochemical, autoimmune, genetic, and histological profiles (Table 1). The laboratory data were assessed by conventional methods. The diagnosis of AIH was established according to the International Autoimmune Hepatitis Group criteria and the Roussel Uclaf Causality Assessment Method (RUCAM) for drug-induced hepatotoxicity was applied to patients with a “probable” or “less likely” diagnosis of AIH.

The authors declare that the procedures were followed according to the regulations established by the Clinical Research and Ethics Committee and to the Helsinki Declaration of the World Medical Association.


**Case Report 1**


A 36-year-old Caucasian woman presented to the hepatology outpatient clinic with asymptomatic transaminase elevation in March 2021. She had been followed up by a hepatologist since June 2018, when she had an isolated episode of fever and diarrhea, accompanied by transaminase elevation; on that occasion, the autoimmunity and virological screening results were negative. She became pregnant in December 2018 and the laboratory exams during and after pregnancy were all negative.

Her laboratory exams in March 2021, performed 7 days after the administration of the first dose of viral vector vaccine (AstraZeneca), showed mild transaminase elevation (aspartate-aminotransferase [AST] 2.2 × Upper Limit of Normal [ULN], alanine-aminotransferase [ALT] 3.5 × ULN, GGT [gamma-glutamyl transferase] 0.3 × ULN). She received her second dose of viral vector vaccine (AstraZeneca) in May 2021 and the transaminase levels at the subsequent follow-up progressively increased. She reported modest alcohol consumption until August 2021 and denied the intake of other potentially hepatotoxic substances. She had a positive family history for immune-mediated and liver diseases, with her sister affected by celiac disease and her grandmother dead at the age of 69 due to hepatocellular carcinoma.

Blood tests in October 2021 showed a significant increase in transaminases (AST 40.5 × ULN, ALT 52.2 × ULN, GGT 2.1 × ULN, ALP [alkaline phosphatase] 2.6 × ULN) and gamma-globulins (2.5 g/dL). HAV, HBV, HCV, CMV, EBV, and HIV serology were negative. Autoimmunity panel was performed: anti-nuclear antibodies (ANA), anti-mitochondrial antibodies (AMA), anti-double-stranded-DNA antibodies (anti-dsDNA), anti-liver-kidney microsomal antibodies (anti-LKM), anti-soluble liver antigen/liver pancreas antibodies (anti-SLA/LP), anti-neutrophil-cytoplasmic antibodies (ANCA), and anti-F-actin antibodies were all negative, whereas anti-smooth muscle antibodies (ASMA) were positive (1:160). Abdominal ultrasound showed normal liver size and parenchymal structure, no focal lesions, non-dilated biliary tract, and no signs of portal hypertension. Hepatic stiffness measured using fibroscan^®^ was 11 kPa. On liver biopsy, rosettes, moderate focal piecemeal necrosis (Figure 1) and mild periportal fibrosis were observed with a moderate-to-severe chronic inflammatory portal infiltrate of plasma cells (Figure 2) and eosinophils. Centrilobular necrosis (Figure 3) and minimal focal bile duct neogenesis were also present. According to the revised original scoring system for diagnosis of autoimmune hepatitis, histology was compatible with AIH (Table 2). HLA testing showed DRB1*07 and DRB1*11.

A definite diagnosis of AIH was made based on the AIH original pre-treatment score of 19, and prednisone 50 mg/day was started, obtaining a rapid reduction of transaminase and gamma-globulin levels. Prednisone was then tapered to 2.5 mg/day, the minimal dose to maintain disease control. The third vaccine dose was temporarily suspended.


**Case Report 2**


A 65-year-old Caucasian woman was admitted to the hospital in August 2021 because of dysuria, severe asthenia, and transaminase elevation. Symptoms presented about one month after the second dose of mRNA vaccine (Moderna^®^), administered in May 2021. Her medical history included diabetes mellitus, deep venous thrombosis of the right leg, cerebral ischemia, surgical removal of a uterine fibroma, cholelithiasis, and past transient positivity for thyroid autoantibodies. The patient denied smoking, alcohol use, and hepatotoxic drugs or herbal supplements.

Her laboratory exams showed elevation of transaminases (AST 8.6 × ULN, ALT 8.6 × ULN), GGT (5.2 × ULN), and gamma globulins (2.5 g/dL), which were normal in March 2021 before vaccination. Serology for HAV, HBV, HCV, CMV, EBV, and HIV infection was negative. Autoimmunity testing showed positivity for ANA (1:160), ASMA (1:160), and ANCA; AMA, anti-dsDNA, anti-LKM, anti-LC1, anti-F-actin, and anti-SLA/LP antibodies were negative. Abdominal ultrasound showed normal liver size and echostructure, no focal lesions, non-dilated biliary tract, and no signs of portal hypertension. Hepatic stiffness measured using Fibroscan^®^ (Echosense, France) was 7.22 kPa. At genetic testing, HLA-DRB1*03 and *11 were found. She refused liver biopsy.

Treatment with prednisone 50 mg/day was started in August and tapered to 12.5 mg/day. Azathioprine 50 mg/day was added in December 2021, obtaining normalization of transaminase, GGT, and gamma-globulin levels. The third vaccine dose was suspended. Prednisone was gradually tapered to 2.5 mg/day in December 2022. Then, the patient autonomously suspended azathioprine treatment and had a mild increase in transaminase levels, which was treated with prednisone 5 mg/day and azathioprine 100 mg/day with biochemical remission at three months.


**Case Report 3**


A 14-year-old Caucasian child with Noonan syndrome presented with asymptomatic transaminase elevation (AST 3.4 × ULN, ALT 5.4 × ULN) at the blood exams in October 2021, two weeks after the second dose of mRNA COVID-19 vaccine (Pfizer). His liver tests were normal in March 2021. Virological screening for HAV, HBV, HCV, CMV, EBV, and HIV infection results were negative, and the patient’s parents reported that he did not intake potentially toxic drugs. The following blood tests confirmed transaminase alterations, up to AST 26.2 × ULN, ALT 31.9 × ULN, GGT 2.3 × ULN, ALP 1.7 × ULN, total bilirubin 5.09 mg/dL (direct bilirubin 2.83 mg/dL), and gamma-globulins 2.0 g/dL. The autoimmunity panel was positive for ANA (1:160) and anti-SMA (1:320), and negative for AMA, anti-dsDNA, ANCA, anti-LC1, anti-F-actin, anti-SLA/LP, and anti-LKM antibodies. An abdominal ultrasound performed in February 2022 showed hepatosplenomegaly, a new finding compared to imaging from April 2021. Accordingly, the patient was sent to our attention in March 2022 to undergo a liver biopsy. Histology documented moderate chronic inflammatory portal infiltrate of plasma cells and eosinophils, along with mild focal piecemeal and centrilobular necrosis; moreover, moderate portal fibrosis, ductal metaplasi, a and PAS-positive macrophages were observed. Histology was therefore compatible with AIH (Table 2). The subsequent magnetic resonance cholangiopancreatography excluded the diagnosis of primary sclerosing cholangitis and other biliary tree alterations. HLA testing showed DRB1*03 and DRB1*07.

Based on the original AIH score of 15, the pre-treatment diagnosis of AIH was probable and Methylprednisolone 48 mg/day was started. The patient had a good response, with normalization of transaminases and cholestasis markers after five weeks. The steroid was then tapered without disease relapses.


**Case Report 4**


A 44-year-old Caucasian woman presenting with acute hepatitis was admitted to the Internal Medicine department in March 2022. Her medical history included recurrent episodes of pneumonia and urinary tract infection. Transaminase levels were normal at her last laboratory exams in March 2021. She received the two doses of mRNA COVID-19 vaccine (Moderna) in April and May 2021, and the booster dose in December 2021. After the booster dose, she persistently suffered pruritus and asthenia, but she did not refer to medical attention. In February 2022, she had symptomatic SARS-CoV-2 infection with fever and asthenia. After three days, pruritus worsened, and after 15 days, she also developed jaundice. As blood tests showed elevated transaminases (AST 50.9 × ULN, ALT 30.3 × ULN), markers of cholestasis (total bilirubin 13 mg/dL, GGT 3.0 × ULN), and gamma-globulins (2.4 g/dL), in March 2022 she was admitted to the hospital. Serology for HAV, HBV, HCV, CMV, EBV, and HIV was negative. The autoimmunity panel showed positivity for ANA (1:160) and anti-LC1 antibodies. No pathological findings were observed on abdominal ultrasound. She was discharged on steroid therapy (prednisone 10 mg/day) with the diagnosis of cholestatic jaundice.

She underwent our first hepatological examination in April 2022, when ALT was only 1.2 × ULN. She performed further examinations in May, which showed mild worsening of liver enzymes (AST 3.2 × ULN, ALT 3.9 × ULN), positivity for ANA (1:80) and anti-SLA/LP antibodies, and negativity for ANCA, AMA, ASMA, anti-LKM, anti-F-actin, and anti-LC1 antibodies. HLA testing showed DRB1*13 and DRB1*14. Abdominal ultrasound was normal and hepatic stiffness measured using elastometry was 9.1 kPa. Liver biopsy showed interface hepatitis with moderate chronic inflammatory infiltrate of lymphocytes and plasma cells, moderate portal fibrosis, and perivenular necrosis; minimal bile duct neogenesis and PAS-positive macrophages were also present. AIH was diagnosed based on an AIH score of 17. Prednisone therapy at the dose of 25 mg/day was started, obtaining transaminase normalization and then tapered to 2.5 mg/day without disease relapses.


**Case Report 5**


A 61-year-old Caucasian man developed asymptomatic hypertransaminasemia (ALT 3.05 × ULN) in July 2021, 42 days after the second dose of mRNA COVID-19 vaccine (Pfizer). The first dose was administered in May 2021. The last liver biochemical profile of July 2020 was normal. He was affected by rosacea, hypercholesterolemia, and psoriasis; his family history was positive for hypothyroidism. He denied toxic drug and herbal intake and serology for HAV, HBV, HCV, CMV, EBV, and HIV was negative; abdominal ultrasound showed liver steatosis without other significant findings, and elastometry was 6.8 kPa.

His liver enzymes were monitored until December 2021 when he was admitted to the hospital due to worsening liver damage and jaundice (AST 22 × ULN, ALT 25.6 × ULN, total bilirubin 10 mg/dL). The autoimmunity panel showed positivity for ANA (1:80) and pANCA; gamma-globulins 1.31 g/dL. HLA testing was positive for DRB1*07 and DRB1*11. Liver biopsy documented the presence of moderate chronic inflammatory infiltrate of lymphocytes, plasma cells and PAS-positive macrophages, moderate interface hepatitis and centrilobular necrosis, and mild periportal fibrosis (Table 2). Immunohistochemistry for CMV and HSV 1,2 was negative.

AIH was diagnosed based on a pre-treatment AIH score of 16. Prednisone 50 mg/day was started, with normalization of liver enzymes and then tapered to 10 mg/day. In February 2022, a booster dose of mRNA vaccine was administered under steroid treatment without AIH reactivation. He autonomously suspended steroid therapy in April 2022 and underwent disease relapse with ALT 11 × ULN. Prednisone 75 mg/day was started and, after transaminase re-normalization, was tapered to 7.5 mg/day.


**Case Report 6**


A 44-year-old Caucasian man with a known diagnosis of AIH came to our attention in November 2021 because of abnormal blood tests after COVID-19 vaccination.

He was diagnosed with AIH in March 2018 after the finding of positive ANA and SMA, and typical liver histology; he started treatment with steroid therapy, obtaining a complete response. He had a drug-induced liver injury due to the consumption of homeopathic supplements, with normalization of transaminases and markers of cholestasis after their suspension. During the follow-up, he was found to carry HLA-DRB1*03 and HLA-B*08 and diagnosed with seronegative arthropathy. Since September 2019, he had been on maintenance therapy with azathioprine only and his exams were normal until March 2021. He had an asymptomatic SARS-CoV2 infection in May 2021 without sequelae.

He received the first dose of mRNA COVID-19 vaccine (Pfizer) in October 2021 and after 15 days, he started complaining of asthenia and altered bowel habits. In November 2021, his laboratory exams showed elevation of transaminases (AST 3.5 × ULN, ALT 8.4 × ULN) andGGT (1.6 × ULN), neutropenia (1830/mmc), and lymphopenia (1200/mmc). Virological screening for HAV, HBV, HCV, CMV, EBV, and HIV infection was negative. He underwent abdominal ultrasound, which revealed a focal lesion consistent with focal nodular hyperplasia whilst displaying no signs of portal hypertension, thrombosis or fibrosis. Laboratory tests spontaneously returned to normal within two months, with the persistence of leukopenia alone.


**Case Report 7**


A 46-year-old Caucasian man with symptoms and signs of acute hepatitis was admitted to the hospital in April 2021. He had a longstanding history of mild GGT and ALT elevation due to unknown causes, which normalized after dietary correction. Moreover, he had a family history of chronic liver disease, as his brother was in follow-up for probable non-progressive familial intrahepatic cholestasis.

In April 2021, the patient contracted SARS-CoV-2 infection, which manifested as interstitial pneumonia and acute hepatitis (ALT 12.5 × ULN, GGT 12 × ULN). He was admitted to the Infectious Diseases Unit for 15 days, and one month after discharge his blood tests normalized without steroidal treatment. In July 2021 he received the first dose of mRNA COVID-19 vaccine (Moderna) and experienced fever, headache, arthralgia and asthenia for the next four days, with spontaneous resolution; on that occasion, he did have laboratory tests performed.

In January 2022, the day after the booster dose of mRNA COVID-19 vaccine (Moderna), the patient came into the emergency department because of a high fever. His laboratory exams revealed transaminase elevation (ALT 7.8 × ULN), whereas gamma-globulins were in range. The virological screening was negative for HAV, HBV, HCV, CMV, EBV, and HIV infection, and autoimmunity testing was negative for ANA, AMA, ASMA, ANCA, anti-LKM, anti-dsDNA, anti-F-actin, anti-SLA/LP, and anti-LC1 antibodies. Abdominal ultrasound showed no pathological alterations. Liver histology was not typical for autoimmune disease, characterized by the presence of PAS-positive macrophages only, in the absence of inflammatory infiltrate, fibrosis, and piecemeal necrosis (Table 2). The RUCAM score was nine, indicating a highly probable causality link between the liver injury and the adverse drug reaction. Finally, he was found to carry HLA-DRB1*11 and HLA-DRB1*15.

His transaminases returned to normal in February 2022 without the need for medications. He is currently in hepatological follow-up.


**Case Report 8**


A 65-year-old Caucasian woman was referred to our institution in August 2021 because of pain in the right hypochondrium and hyperchromic urine in the last month. Her medical history included goiter thyroidectomy currently on thyroxine replacement therapy. She denied taking any supplements or herbal remedies. She had a history of clinical SARS-CoV2 infection in March 2020 and she received the second dose of mRNA COVID-19 vaccine (Moderna) in July 2021.

In August 2021, her physical examination was normal. Laboratory exams showed an increase both in transaminases (AST 3.3 × ULN, ALT 3.5 × ULN) and markers of cholestasis (GGT 5.5 × ULN, ALP 4.1 × ULN). Bilirubin levels and gamma-globulins were in range. Serological tests for HAV, HBV, HCV, CMV, EBV, and HIV were negative for active infection. Testing for autoimmune liver serology (ANA, ASMA, AMA, anti-dsDNA, ANCA, anti-LKM, anti-SLA/LP, anti-F-actin, and anti-LC1 antibodies) was negative. Abdominal ultrasound and magnetic resonance cholangiopancreatography only detected mild steatosis and excluded any alterations to the liver, gallbladder, biliary tract, and pancreas. Upon liver biopsy, a non-specific chronic inflammatory infiltrate in the portal area, with plasma cells and eosinophils, and mild piecemeal necrosis was observed, in the absence of fibrosis and cholestasis. (Table 2). HLA testing showed the presence of DRB1*11. The RUCAM score was eight, indicating a probable causality link between the liver injury and the adverse drug reaction.

She did not undergo steroidal treatment and her liver biochemistry tests returned to normal after five months. The third vaccine dose was suspended.

## 3. Discussion

In this case series, we presented six cases of AIH and two cases of liver biochemistry test elevation following COVID-19 vaccination (seven cases after mRNA vaccine and one after viral vector vaccine) or SARS-CoV-2 infection (patient 4), ranging from asymptomatic transaminase elevation to severe acute hepatitis requiring hospitalization. Despite the global vaccination campaign, rare cases of AIH have been reported in the literature. Given the rarity of the event, the association cannot be established with certainty, but the similarities among these cases and the temporal relationship with the exposure to the trigger could support this hypothesis.

The potential mechanism for this association can be explained by the theory of molecular mimicry between viral antigens and self-antigens in the liver, as demonstrated for other viruses [40]. Indeed, Vojdani et al. [41] demonstrated in an in vitro study that the spike protein S1 of SARS-CoV-2 shares structural similarities with some human tissue proteins (transglutaminase 3, transglutaminase 2, anti-extractable nuclear antigen, nuclear antigen, and myelin basic protein) expressed in a variety of tissues. As this viral protein is the target of the antibodies produced by the immune system after exposure to the vaccine or infection [42], a cross-reaction with human proteins is possible. Moreover, other authors have suggested a correlation between liver injury and the systemic inflammation induced by SARS-CoV-2, which can cause direct damage to the inflammasome pathways in the hepatocytes, contributing to cell death [43].

The time to symptoms onset, in relation to the vaccination or the infection, varied from two to six weeks in most cases, in line with the data reported by other authors [6,12,13,15,18,25]. A shorter latency period has been observed in two of the reported cases (cases 1 and 4), probably because these patients had been recently sensitized to the pathogen and the subsequent exposure to infection or vaccination caused a faster immune reaction. All but patient 1 had a personal or familial history of autoimmune diseases, and this feature increases the probability for autoimmune events, such as AIH, after a trigger as a vaccination or an infection [29].

A liver biopsy was performed in six patients, and four of them (cases 1, 3, 4, and 5) were compatible with the diagnosis of AIH according to the simplified criteria for AIH [39]. The histology of the patients with new-onset AIH displayed one or more features typical of the disease, including interface hepatitis and hepatic rosettes. Three of them showed concomitant ductular reactions, which is not a typical feature of AIH but is known to occur as a consequence of the damage caused by severe and prolonged inflammation [44]. The presence of PAS-positive macrophages and eosinophils are other non-specific findings that can be present in AIH [44].

Regarding the genetic factors, alleles known to confer susceptibility to AIH in different populations [29] were found in all reported AIH patients: HLA-DRB1*03 was positive in cases 2, 3, and 6 (3/6 patients), HLA-DRB1*07 in cases. 1, 3, and 5 (3/6 patients), and HLA-DRB1*13 and HLA-DRB1*14 in case 4 (1/6 patients). HLA-DRB1*15, described in some AIH non-Caucasian patients [29], was present in case 7 (acute hepatitis without AIH). A peculiar finding of our study is the presence of HLA DRB1*11 in three out of six patients with AIH (patients 1, 2, and 5), which is concordant with the observation of Ghielmetti et al. [11] in their case report. Although the role of HLA DRB1*11 in liver diseases has been poorly investigated, in the available literature, this allele has been significantly associated with resistance to chronic HBV and HCV infection in Chinese, Brazilian, and European populations [45,46], and with a weak protection to AIH in studies conducted on Iranian and Iraqi groups of patients [47,48]. This may suggest that certain allelic variants play an important role in the immune response against SARS-CoV-2 infection or in AIH pathogenesis, but further data and investigations in the European population are needed to identify an association. Concerning the peculiar case of patient 3, a genetic predisposition to AIH may be present in patients with Noonan syndrome, as this is known to be associated with other autoimmune diseases [49].

All the patients with a new diagnosis of AIH were treated with steroids with subsequent normalization of their liver biochemical profiles. The response to immunosuppression further validates the hypothesis of an underlying immune reaction. Since fibrosis and chronic inflammation were present on their histology, we can hypothesize that they had a quiescent AIH or a predisposition to AIH and that, in the absence of other known triggers, the vaccine has made the disease patent. One patient with a previous AIH diagnosis (case 6) had transaminase elevation while on treatment with azathioprine, which spontaneously resolved without changes in therapy. A similar case has been reported by Mahalingham et al. [25], who described a patient transplanted for AIH that was stable for many years on immunosuppressive treatment and had an immune-mediated flare after COVID-19 vaccination. Differently from patient 6, he required add-on steroid therapy, so, at the moment, the most appropriate treatment should be evaluated on a case-by-case basis.

Finally, patients 7 and 8 had acute hepatitis but did not meet the criteria for AIH; their liver tests normalized without the need for immunosuppression and they showed HLA DRB1*11, without an HLA strongly associated with AIH in Caucasians (i.e., HLA DRB1*03, HLA DRB1*07, HLA DRB1*13 and HLA DRB1*14). A similar case was described by Izagirre et al. [50], supporting the hypothesis that HLA DRB1*11 could be associated with acute hepatitis self-resolving without therapy when dissociated by AIH-predisposing HLA alleles.

The heterogeneity observed in our group can be made clear by considering the definition of drug-induced autoimmune liver disease: indeed, this term encompasses the three entities of drug-induced-AIH (DI-AIH), AIH with drug-induced liver injury (DILI), and immune-mediated DILI (IM-DILI) [38]. Unfortunately, there are neither histological nor clinical features that allow for a definite distinction between AIH and DILI. The presence of advanced fibrosis or chronic inflammation on histology is suggestive of a long-standing process, which is more typical of pre-existing AIH with superimposed DILI, and DI-AIH rather than IM-DILI. In contrast to patients with pre-existing AIH, patients with DI-AIH usually have a low-grade disease or even just a predisposition to AIH that is awakened by a new drug [38]. IM-DILI is a form of autoimmune hypersensitivity that frequently resolves with drug cessation without relapses [38]. In clinical practice, distinguishing DI-AIH and AIH-DILI from IM-DILI has an important implication, as in the former, immunosuppression is essential to avoid relapses, whereas in the latter, immunosuppression may not be required and, when needed, it can be discontinued after response to treatment [27,38]. 

While this study yields valuable insights, several limitations should be acknowledged. The relatively small sample size restricts our capacity to establish definitive conclusions. The absence of a control group for HLA type comparison and the inability to ascertain causality between vaccination/infection and liver disease are also noteworthy constraints. Additionally, the single-center nature of this experience may limit the generalizability of our findings. Furthermore, the potential for reporting bias following wide spread vaccination campaigns should be considered.

## 4. Conclusions

In conclusion, while a causal link between autoimmune hepatitis and the vaccine cannot be demonstrated, this association could be more than coincidental due to the temporal relationship with the exposure to the triggering factor, the similarities in clinical and laboratory features, and the response to immunosuppression. On the other hand, we must underline that the onset or flare of AIH after vaccination is a rare adverse event [38] and the benefits from global vaccination far outweigh the risk. The number of cases reported is very few, so vaccination does not pose a more prominent risk of acute hepatitis than natural infection itself. Pandemic immunization demonstrated to be safe and effective for COVID-19 containment and vaccine refusal cannot be conceived.

In view of the new doses of the COVID-19 vaccine, clinicians should include AIH and DILI among the differential diagnoses for transaminases elevation and eventually consider checking aminotransferases two to four weeks after COVID-19-vaccination in selected patients with clinical or family history of autoimmune diseases.

Regardless of the possible causal role of the vaccine or COVID-19 infection that we hypothesized in a few cases, our study wants to focus the attention on HLA predisposition in acute hepatitis and AIH, and to highlight the need for further studies that could better identify the role of HLA DRB1*11 in the pathogenesis of liver injury.

## Figures and Tables

**Figure 1 biomedicines-11-02848-f001:**
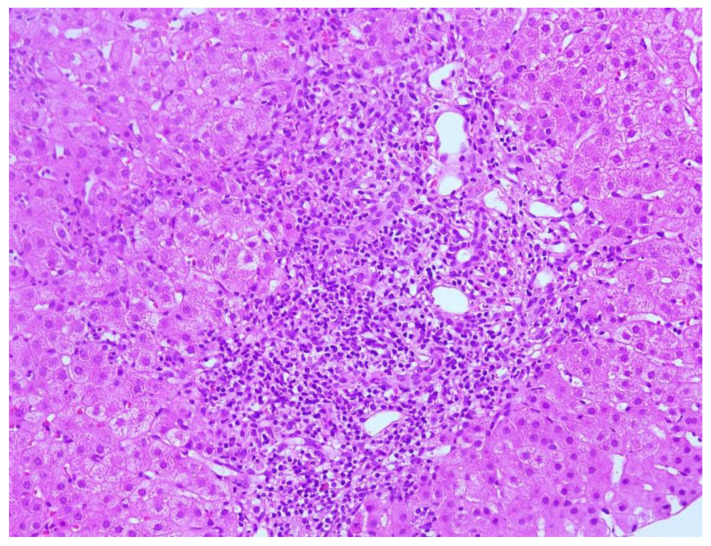
Expanded portal tract: severe portal inflammation with interface activity; Hematoxylin-Eosin (HE), 20X.

**Figure 2 biomedicines-11-02848-f002:**
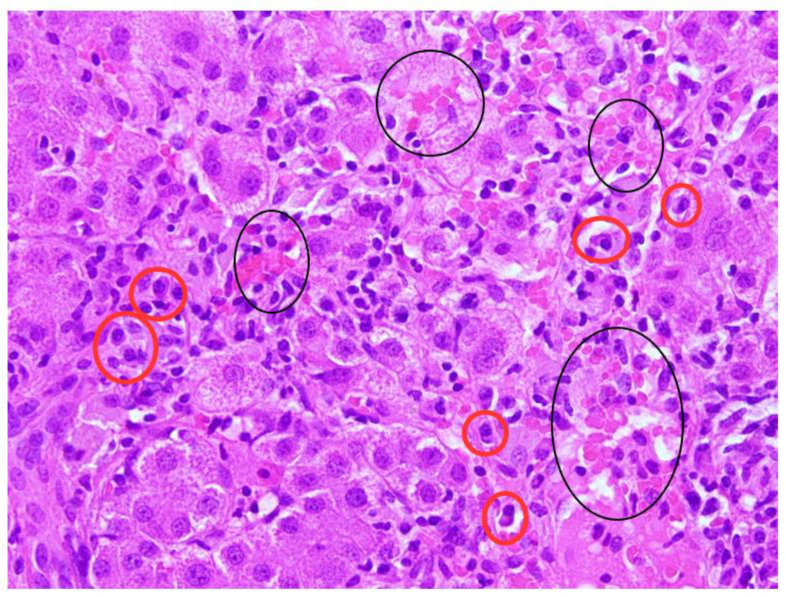
Spotty necrosis, non-specific autoimmune-hepatitis-related element (black circles) in lobular hepatitis, and several plasma cells (red circles); HE, 40X.

**Figure 3 biomedicines-11-02848-f003:**
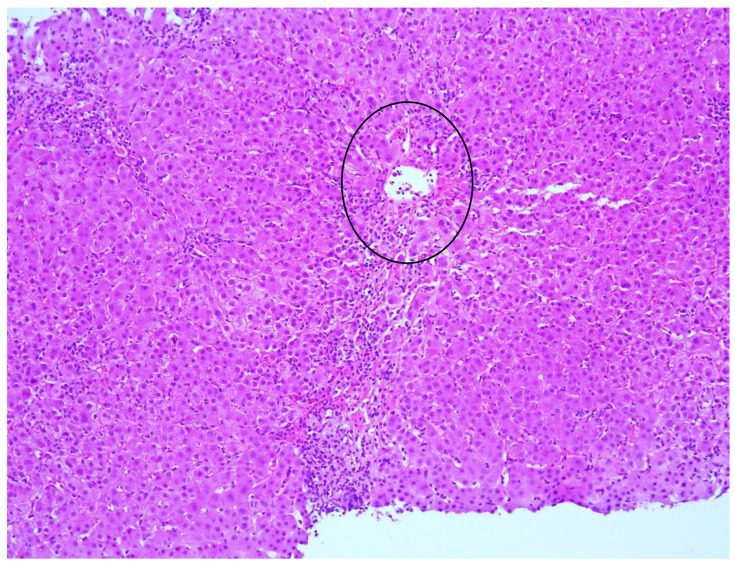
Centrilobular necrosis (black circle), HE, 10X.

**Table 1 biomedicines-11-02848-t001:** Patients presenting with acute hepatitis after COVID-19 vaccine or SARS-CoV-2 infection.

Clinical Characteristics	Vaccine/ Infection	Time to Symptoms Onset (Days)	AST/ALT (U/L) ^a^	ALP/GGT (U/L) ^a^	Auto-Antibodies (+)	IgG (mg/dL) ^a^	Histology for AIH ^b^	AIH Score ^c^	HLA DRB1	Steroid Treatment
1. Female, 36 Asymptomatic	Viral vector vaccine	7	1500/2087	331/113	SMA	2948	Compatible	19	*07 *11	Yes
2. Female, 65 Symptomatic	mRNA vaccine	28	320/344	170/286	ANA, SMA, ANCA	2450	Not performed	-	*03 *11	Yes
3. Male, 14 Asymptomatic	mRNA vaccine	15	970/1276	216/126	ANA, SMA	2167	Compatible	15	*03 *07	Yes
4. Female, 44 Symptomatic	mRNA vaccine Infection	15 3	1885/1211	109/165	ANA, SLA/LP	2180	Compatible	17	*13 *14	Yes
5. Male, 61 Symptomatic	mRNA vaccine	42	818/1024	310/-	ANA, ANCA	1384	Compatible	16	*07 *11	Yes
6. Male, 44 Asymptomatic	mRNA vaccine	15	131/335	151/86	ANA, SMA	-	Typical (2018)	-	*03	No
7. Male, 46 Symptomatic	Infection mRNA vaccine	7 1	-/500 -/311	-/660 -/-	Negative	-	Atypical	3	*11 *15	No
8. Female, 65 Symptomatic	mRNA vaccine	28	141/123	518/303	Negative	-	Atypical	11	*11	No

IgG, immunoglobulin G; ANA, antinuclear antibody; SMA, smooth muscle antibodies; dsDNA, double stranded DNA antibodies; LKM, liver-kidney microsomal antibody; SLA/LP, soluble liver antigen/liver pancreas antibody; ANCA, anti-neutrophil-cytoplasmic antibodies; AZA, azathioprine; UDCA, ursodeoxycholic acid. ^a^ Reference values: AST 1–37 U/L; ALT 1–40 U/L; ALP 38–126 U/L; GGT 1–55 U/L; IgG 800–1800 mg/dL. ^b^ According to the revised original scoring system for diagnosis of autoimmune hepatitis [39]. ^c^ According to the International Autoimmune Hepatitis Group criteria for diagnosis of AIH [39].

**Table 2 biomedicines-11-02848-t002:** Patients’ liver histology findings.

Clinical Characteristics	Interface Hepatitis (Figure 1)	Lym	Plasmacells (Figure 2)	Eos	PAS-pos MP	Centrilobular Necrosis (Figure 3)	Emperipolesis	Rosettes	Ductular Reaction	Fibrosis	AIH Score
1. Female, 36	Moderate, focal	Yes	Moderate	Yes	No	Yes	No	Yes	Minimal bile ducts neogenesis	Yes	19
3. Male, 14	Mild, focal	Yes	Mild	Yes	Yes	Yes	No	No	Bile ducts neogenesis, ductal metaplasia	Yes	15
4. Female, 44	Moderate, focal	Yes	Mild	No	Yes	Yes	No	No	Minimal bile ducts neogenesis	Yes	17
5. Male, 61	Moderate	Yes	Moderate	Yes	Yes	Yes	No	No	No	Yes	16
7. Male, 46	No	No	No	No	Yes	No	No	No	No	No	3
8. Female, 65	Mild, focal	Yes	Few	Yes	Yes	No	No	No	No	No	11

Lym, lymphocytes; Eos, eosinophils; PAS-pos MP, Periodic Acid–Schiff-positive macrophages.

## Data Availability

All data analyzed during this study are included in this article.

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
