# Peer review of "Investigating Acute Hepatitis after SARS-CoV-2 Vaccination or Infection: A Genetic Case Series"

_biomedicines, 2023, doi:10.3390/biomedicines11102848_

Round 1
Reviewer 1 Report
The authors reported 8 patients who developed acute hepatitis, 7 of them possibly associated with COVID-19 vaccination. They analyzed the genetic contribution to this clinical manifestation. Some suggestions are included to improve the manuscript.
1. The title and the first sentence of the Abstract are misleading. This study is a report of 8 cases, not a review on the eventual association of COVID-19 vaccine and acute hepatitis. In the Abstract, the term ¨many¨ in ¨many cases¨ is also misleading. The frequency of autoimmune adverse effect is really low. The extraordinary benefits of counting with diverse vaccines against COVID-19 is not mentioned in this manuscript.
2. Introduction. First sentence. COVID-19 vaccines have been associated in rare occasions with autoimmune pathologies. It is important to stress in all this manuscript that these associations are really infrequent.
3. Results. A brief description of the 8 patients (6 with AIH) and their HLA results should be included in Results, not only showing a table. The frequency of HLA DRB1 for each allele is only described in Discussion but not in results. The frequencies for each allele are not presented clearly. Thus the association with HLA alleles and AIH is not clearly presented.
4. Line 301: our patients?
5. Table 1. All but one of the 7 patients developing AIH or related clinics after vaccination exhibited auto-antibodies, suggestive of autoimmune phenomena previous to vaccination. This is not mentioned. What is the probability of AIH manifestations in a population with auto-antibodies? This should be addressed with more details. Were these patients analyzed previously for hepatic enzymes, to discard periodic flare of these enzymes, due to the autoimmune condition? Were these patients studied after vaccination against another pathogen, for example Influenza, to know if hepatic enzymes were also elevated? This should be discussed and presented with more details.
6. Line 344-348 are fundamental and just mentioned in few lines in this manuscript. This should be stressed in Abstract.
7. In general, the authors should stress that the adverse effects presented in this study, as in previous ones, is really an exception of an extraordinary advent: the availability of safe vaccines for controlling a pandemic.
Author Response
Dear reviewer,
we thank you for the observations and comments you have provided. Following the instructions you provided, we have made some changes to the text to make it more fluent. Below are the responses to your questions:
Question 1:
The title and the first sentence of the Abstract are misleading. This study is a report of 8 cases, not a review on the eventual association of COVID-19 vaccine and acute hepatitis.
Answer: We changed the title and the first phrase of the Abstract following your indications
Question 2:
In the Abstract, the term ¨many¨ in ¨many cases¨ is also misleading
Answer: We fixed these mistakes.
Question 3:
The frequency of autoimmune adverse effect is really low. The extraordinary benefits of counting with diverse vaccines against COVID-19 is not mentioned in this manuscript
Answer: We fixed this concept, see lines 416-421
Question 4:
Introduction. First sentence. COVID-19 vaccines have been associated in rare occasions with autoimmune pathologies. It is important to stress in all this manuscript that these associations are really infrequent.
Answer: We fixed the first sentence of introduction and we stressed in all the manuscript that the association are infrequent. See lines 43-45, 321-323, 418-419, 427
Question 5:
Results. A brief description of the 8 patients (6 with AIH) and their HLA results should be included in Results, not only showing a table. The frequency of HLA DRB1 for each allele is only described in Discussion but not in results. The frequencies for each allele are not presented clearly. Thus the association with HLA alleles and AIH is not clearly presented.
Answer: HLA results are included in description of every patient (lines 119, 163, 190, 215, 235-236, 254, 291, 312). In discussion, we analyzed the frequencies for each HLA allele in our AIH patients at lines 353-358
Question 6:
Line 301: our patients?
Answer: We fixed it
Question 7:
Table 1. All but one of the 7 patients developing AIH or related clinics after vaccination exhibited auto-antibodies, suggestive of autoimmune phenomena previous to vaccination. This is not mentioned. What is the probability of AIH manifestations in a population with auto-antibodies? This should be addressed with more details. Were these patients analyzed previously for hepatic enzymes, to discard periodic flare of these enzymes, due to the autoimmune condition? Were these patients studied after vaccination against another pathogen, for example Influenza, to know if hepatic enzymes were also elevated? This should be discussed and presented with more details.
Answer: Our patients didn’t have history of transaminases alterations, so they had not tested for autoimmunity in the past: guidelines suggest dosing autoantibodies only when clinical suspicion is present. In Results, we report last transaminases dosage before Covid-19 vaccination (lines 92-93, 156-157, 175, 199, 226-227, 255-256, 270). All patients administered other vaccinations in the past, but they denied clinical or biochemistry signs suggestive for hepatitis.
All our patients with a new diagnosis of AIH (no. 1,2,3,4,5) did not have a history of persistent transaminases alterations, so we do not know if their autoantibodies were present before Covid-19 vaccination because no one had reasons for dosing them: guidelines suggest searching autoantibodies only when clinical suspicion is present because they are not specific and can be present also for other hepatic and non hepatic diseases. Only patient no.1 showed transaminases modifications in 2018 during gastrointestinal infection with diarrhea and fever, then normalized without reactivations: on that occasion, autoimmunity was negative. ANA and SMA are markers of AIH-1, which account for about 75% of patients, but are not disease specific and show a wide range of heterogeneity in terms of antigenic specificity".
Question 8:
Line 344-348 are fundamental and just mentioned in few lines in this manuscript. This should be stressed in Abstract.
Answer: We fixed it also in abstract. See lines 29-33
Question 9:
In general, the authors should stress that the adverse effects presented in this study, as in previous ones, is really an exception of an extraordinary advent: the availability of safe vaccines for controlling a pandemic.
Answer: We fixed that. In abstract (line 15) starts introducing this concepts. At lines 416-421 we conclude the adverse events of the study are rare and vaccination safety outweighs risks.
Reviewer 2 Report
The study by Bernasconi E. et al. is a Case Report study describing 8 cases of patients who experienced acute hepatitis related to SARS-CoV2: 5 with autoimmune hepatitis (AIH) after SARS-CoV2 vaccination, 1 AIH relapse after vaccination, and 2 non-autoimmune hepatitis.
Major comments:
· General comment. Authors should consider their study as a “Case Report” because the low number of included patients does not allow for statistical analysis or conclusions. Therefore this “description” of 8 cases should be reconsidered for biomedicines with a “Case Report” format
· Title. The title should include important words for a better description of the study. Please include words such as “after SARS-COV2 vaccination” and “Case Report”
· Abstract. Please, describe the method to evaluate the cases (see “methods section”).
· Patients and Methods. Please, include the inclusion and exclusion criteria for selecting the cases. Please, include the number of the Ethics Committee. Authors could follow a guideline for a better description of their cases. (i.e.: http://dx.doi.org/10.1016/j.jclinepi.2017.04.026).
· Case Reports. Please include Table 1 and Table 2 as part of the case description.
· Table 1. Please include RUCAM in those patients with DILI suspicion. Please, include information regarding prednisone or immunosuppressive treatment, response, and follow-up.
· Table 2. Please, include the AIH score
· Discussion. Please discuss the differences between DILI and AIH-like syndromes.
Author Response
Dear reviewer,
taking this opportunity to thank you for your observations and comments, which have been very valuable in improving the quality of the text, we are providing you with the answers to your questions. Moreover, we would like to inform you that some modifications have been made to the text with the aim of making it more understandable.
Question 1: General comment. Authors should consider their study as a “Case Report” because the low number of included patients does not allow for statistical analysis or conclusions. Therefore this “description” of 8 cases should be reconsidered for biomedicines with a “Case Report” format
Answer: We thank you for puntualization. Current format has been accepted by editor, and modification in 'Case Report' format requires too much time, considering short time available for major revision
Question 2: Title. The title should include important words for a better description of the study. Please include words such as “after SARS-COV2 vaccination” and “Case Report”
Answer: We changed the title following your indication. We used the words “Case series” instead of “Case report”, since this paper includes 8 cases.
Question 3: Abstract. Please, describe the method to evaluate the cases (see “methods section”).
Answer: We added a few sentences from line 20 to 25 to describe the methods in the abstract.
Question 4: Patients and Methods. Please, include the inclusion and exclusion criteria for selecting the cases.
Answer: We described inclusion and exclusion criteria in the “Patients and Methods” section and further detailed the methods used to evaluate the patients (see lines 68-76).
Question 5: Please, include the number of the Ethics Committee. Authors could follow a guideline for a better description of their cases. (i.e.: http://dx.doi.org/10.1016/j.jclinepi.2017.04.026).
Answer: Ethics committee approval was waived for this study because it does not request authorization for case series including less than ten patients.
Question 6: Case Reports. Please include Table 1 and Table 2 as part of the case description.
Answer: We added a full description of the histological findings in lines 114-119, 184-188, 217-220, 236-238, 286-289, 309-311. Information in Table 1 was already included in the case description.
Question 7: Table 1. Please include RUCAM in those patients with DILI suspicion. Please, include information regarding prednisone or immunosuppressive treatment, response, and follow-up. Table 2. Please, include the AIH score.
Answer: We included the RUCAM score in the description of case number 7 and case number 8 (lines 289-290, 312-313); we did not include them in the table since this score was applied to two patients only, in whom an autoimmune etiology was excluded.
Information regarding prednisone-immunosuppressive treatment has been included in the case description (lines 143-145, 164-165, 167-170, 192, 209-210, 220-222, 240-241, 243-245).
We added a column with the AIH score to Table 1, following your suggestion.
Question 8: Discussion. Please discuss the differences between DILI and AIH-like syndromes.
Answer: Thank you for this comment, we added lines 394-404 to clarify this concept.
Reviewer 3 Report
This paper describes 8 cases of acute hepatitis occurring after COVID-19 vaccination or SARS-CoV-2 infection. The cases are interesting and provide clinical details that could aid other clinicians in diagnosis and management of similar patients. It is based on real clinical cases with detailed descriptions of their history, laboratory tests, histology and treatment outcomes. Reporting rare but serious adverse events is important. While the number of cases is small, the authors point out that vaccination/infection-related acute hepatitis is likely an uncommon occurrence. The cases highlight the need for further investigation rather than providing definitive evidence. The potential link between vaccination/infection and liver injury is biologically plausible based on mechanisms like molecular mimicry. The temporal association, though not proof, supports the possibility. Identifying HLA-DRB1*11 as a potential genetic risk factor generates a hypothesis to test in larger studies. This finding could be valuable for understanding mechanisms.
= Potential weaknesses that need to be concisely addressed:
- Small number of cases limits ability to draw definitive conclusions
- Lacks control group for comparison of HLA types
- Causality between vaccination/infection and liver disease cannot be proven
- Single center experience may not be generalizable
- Potential for reporting bias of cases after widespread vaccination
none.
Author Response
Dear reviewer,
we thank you for the observations and comments you have provided.
We have incorporated your suggestions regarding the study's limitations, as you can see from lines 405 to 411
Round 2
Reviewer 1 Report
The authors addressed satisfactorely the reviewers comments.
Reviewer 2 Report
The authors have resubmitted a revised version of their manuscript. The revised manuscript and the author’s responses are satisfactory. Therefore their revised manuscript can be accepted in biomedicines